# Structural basis of sodium-dependent bile salt uptake into the liver

Kapil Goutam[1,2], Francesco S. Ielasi[2], Els Pardon[3,4], Jan Steyaert[3,4] & Nicolas Reyes[1,2]✉

The liver takes up bile salts from blood to generate bile, enabling absorption of lipophilic nutrients and excretion of metabolites and drugs[1]. Human Na+– taurocholate co-transporting polypeptide (NTCP) is the main bile salt uptake system in liver. NTCP is also the cellular entry receptor of human hepatitis B and D viruses[2,3] (HBV/HDV), and has emerged as an important target for antiviral drugs[4]. However, the molecular mechanisms underlying NTCP transport and viral receptor functions remain incompletely understood. Here we present cryo-electron microscopy structures of human NTCP in complexes with nanobodies, revealing key conformations of its transport cycle. NTCP undergoes a conformational transition opening a wide transmembrane pore that serves as the transport pathway for bile salts, and exposes key determinant residues for HBV/HDV binding to the outside of the cell. A nanobody that stabilizes pore closure and inward-facing states impairs recognition of the HBV/HDV receptor-binding domain preS1, demonstrating binding selectivity of the viruses for open-to-outside over inward-facing conformations of the NTCP transport cycle. These results provide molecular insights into NTCP 'gated-pore' transport and HBV/HDV receptor recognition mechanisms, and are expected to help with development of liver disease therapies targeting NTCP.

Bile salts are essential molecules for absorption of lipophilic nutrients and vitamins (vitamin A, D, E and K) in the small intestine, as well as for maintenance of endocrine and cholesterol homeostasis and excretion of toxins[1]. The vast majority—more than 90%—of the body's bile salts pool is recycled daily, shuttling between intestine and liver, where bile salts are used to aid nutrient absorption and generate bile, respectively. Human members of the solute carrier 10 (SLC10) protein family are key bile salt transporters for the maintenance of enterohepatic circulation[5,6]: NTCP[7] (also known as SLC10A1) is mainly expressed in the hepatocyte basolateral membrane, and constitutes the main active transport route of bile salts into the liver from blood, whereas apical sodium-dependent bile acid transporter[8] (ASBT (also known as SLA10A2)) is expressed in ileum enterocytes and takes up bile salts from the intestinal lumen. Both transporters are important pharmacological targets, as they can be used to facilitate oral absorption[9,10] (ASBT) and liver uptake[11,12] (NTCP) of drugs conjugated to bile salts, and are involved in the action mechanism (ASBT)[13] and pharmacokinetics (NTCP)[14,15] of cholesterol-lowering therapies. Moreover, NTCP down-regulation in mouse models is associated with increased cholesterol and phospholipid excretion[16], as well as decreased weight gain with a high-fat diet[17]. Notably, NTCP has a fundamental role in liver pathology, as the human cellular entry receptor for HBV/HDV[2,3]. Chronic HBV infection is a major cause of hepatocellular carcinoma and liver cirrhosis, and affects around 250 million people globally[18,19]. The viruses use the myristoylated and unstructured N-terminal domain in the large envelope protein—the preS1 domain (myr-preS1)—to recognize and bind

human NTCP[20–22], explaining viral hepatotropism and the narrow range of animal hosts. Consistently, myristoylated peptides encompassing the residues 2–48 of myr-preS1 (myr-preS1$_{48}$) act as potent inhibitors of HBV/HDV entry into cells[23–26].

Structural insights into the transport mechanism of NTCP and ASBT have come from early X-ray crystal structures of prokaryotic homologues that revealed a ten-transmembrane-helix topology, arranged into core and panel domains[27,28]. The homologues follow an alternating-access transport mechanism, in which relative movements of the two domains provide alternating access to substrate- and sodium-binding sites on opposite sides of the membrane.

Here we set out to study the structural basis of human NTCP function using cryo-electron microscopy (cryo-EM) in combination with conformation-specific nanobodies to reveal key conformational transitions of the NTCP transport cycle.

## Cryo-EM structure determination

Human NTCP is a relatively small (approximately 38 kDa) dynamic membrane protein that lacks soluble folded domains and is biochemically unstable in non-denaturing detergent solution, posing a challenge for single-particle cryo-EM structure determination. To overcome these problems, we first exchanged amino acids in the sequence of wild-type NTCP for consensus residues of representative vertebrate orthologues to confer stability to the protein[29] (Methods). The initial consensus design, NTCP$_{CO}$, was more stable than wild-type NTCP in

[1]Membrane Protein Mechanisms Group, European Institute of Chemistry and Biology, University of Bordeaux, CNRS-UMR5234, Pessac, France. [2]Membrane Protein Mechanisms Unit, Institut Pasteur, Paris, France. [3]Structural Biology Brussels, Vrije Universiteit Brussel, VUB, Brussels, Belgium. [4]VIB-VUB Center for Structural Biology, VIB, Brussels, Belgium. ✉e-mail: nicolas.reyes@u-bordeaux.fr

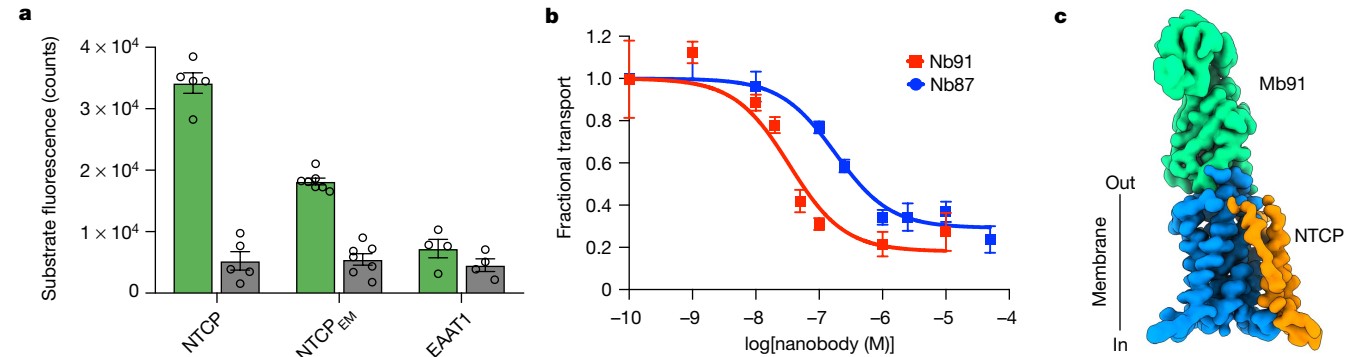

**Fig. 1 | Functional and structural analyses of NTCP_EM–Nb complexes.**
**a**, Uptake of the fluorescent substrate analogue (tauro-nor-THCA-24-DBD) in cells expressing NTCP constructs or negative control (excitatory amino acid transporter EAAT1) using sodium-based (green) and choline-based (grey) buffers, respectively. Bars depict mean of 5 (NTCP), 7 (NTCP_EM) and 4 (EAAT1) biologically independent experiments, error bars represent s.e.m. and circles show values from individual experiments. **b**, Inhibition by Nb91 (red) and Nb87 (blue) of tauro-nor-THCA-24-DBD transport in cells expressing NTCP_EM. Solid lines are fits of a single-site binding equation (Methods). Squares depict mean of three biologically independent experiments and error bars represent s.e.m. **c**, Density corresponding to NTCP_EM core (blue) and panel (orange) domains, and the nanobody part of Mb91 (green).

detergent solution. To minimize the number of consensus exchanges and maximize stability, we determined the contribution of single exchanges, and retained only those that increased stability, yielding a final construct that shares approximately 98% amino acid identity with wild-type NTCP (Extended Data Fig. 1) and enables purification of monodisperse material in milligram amounts. We refer to this construct as NTCP_EM. NTCP_EM showed robust Na⁺-dependent uptake of the fluorescent substrate analogue tauro-nor-THCA-24-DBD ($4.5 \pm 1.3$-fold increase in the sodium- over the choline-based condition), similar to that of wild-type NTCP ($10.2 \pm 3.9$), whereas control cells expressing the unrelated Na⁺-dependent neurotransmitter transporter EAAT1 lacked bile salt uptake ($1.6 \pm 0.1$ sodium-dependent increase) (Fig. 1a). These results show that the transport mechanism is conserved in NTCP_EM.

Second, to provide molecular features on the NTCP_EM surface for cryo-EM analysis, we generated and selected nanobodies that potently bind NTCP_EM. Nanobody (Nb)87 and Nb91 inhibit Na⁺-induced fluorescent-substrate uptake by cells expressing NTCP_EM with half-maximal inhibitory concentrations (IC_50) of approximately 180 and 34 nM (Fig. 1b), respectively, showing that they recognize NTCP_EM from the extracellular side, and suggesting that they stabilize conformational intermediates of the transport cycle. During cryo-EM sample optimization, we screened NTCP_EM complexes with these nanobodies and megabody scaffolds that result in an additional 85 kDa of folded domains[30] in both detergent solutions, as well as reconstituted in nanodiscs. This yielded final cryo-EM maps of NTCP_EM–Nb87 in nanodiscs, and NTCP_EM–megabody (Mb)91 in detergent at overall resolutions of 3.7 and 3.3 Å, respectively, enabling structure determination (Fig. 1c, Extended Data Figs. 2–4, Extended Data Table 1).

## NTCP architecture

NTCP_EM adopts an SLC10 fold with two structurally distinct domains—core and panel (Fig. 2a, b)—and contains nine transmembrane α-helices (TM1–9) with an unstructured N terminus on the extracellular side. The transmembrane helices are connected by short loops, as well as extracellular α-helices (ECH) and intracellular α-helices (ICH) lying nearly parallel to the membrane. The panel domain is formed by TM1, TM5 and TM6, and has lost pseudo-internal symmetry compared with its equivalent in SLC10 prokaryotic homologues, owing to the evolutionary loss of one transmembrane helix. The NTCP_EM core domain is formed by packing of two helix bundles, TM2–4 and TM7–9, which are related by pseudo-two-fold symmetry (Cα root mean squared deviation (r.m.s.d.) ≈ 5 Å). TM3 and TM8 unwind close to the middle of the membrane, and pack against each other to form a characteristic X-shaped structure that displays highly conserved polar residue motifs among vertebrate SLC10 bile salt transporters (Extended Data Figs. 5, 6).

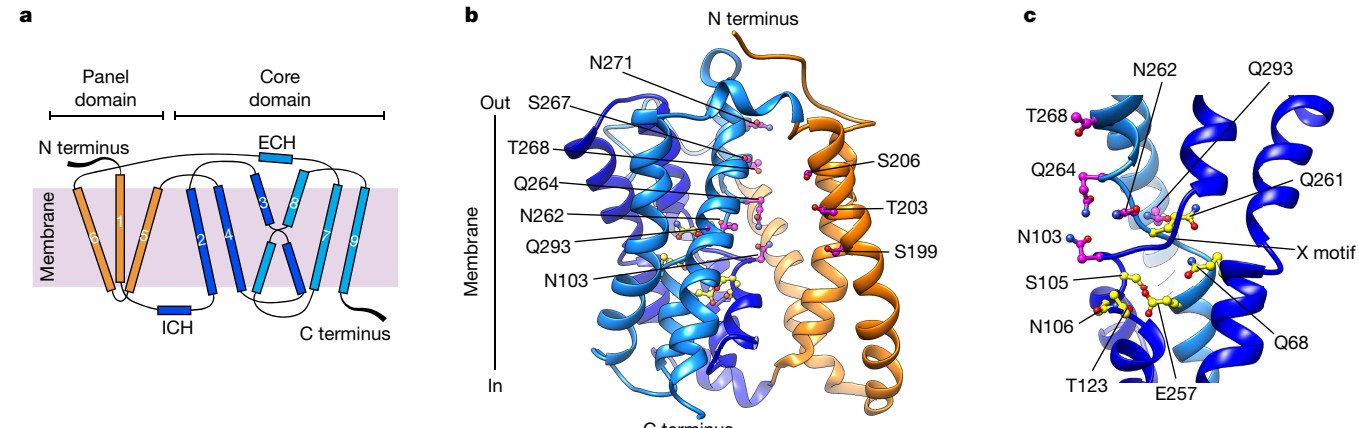

**Fig. 2 | NTCP topology and architecture. a**, Cartoon representation of NTCP topology. **b**, Structure of NTCP_EM in complex with Mb91. Mb91 is omitted for clarity. TM2–TM4 (dark blue) and TM7–TM9 (light blue) in the core domain are related by pseudo-two-fold symmetry, and the panel domain is formed by TM1 and TM5–TM6 (orange). Polar conserved residues lining the space between the core and panel domain (pink), as well as sidechains contributing to Na1 and Na2 (yellow) are shown. **c**, The X motif is formed by unwinding of TM3 and TM8. Only TM2 and TM3 (dark blue) and TM8 (light blue) are shown.

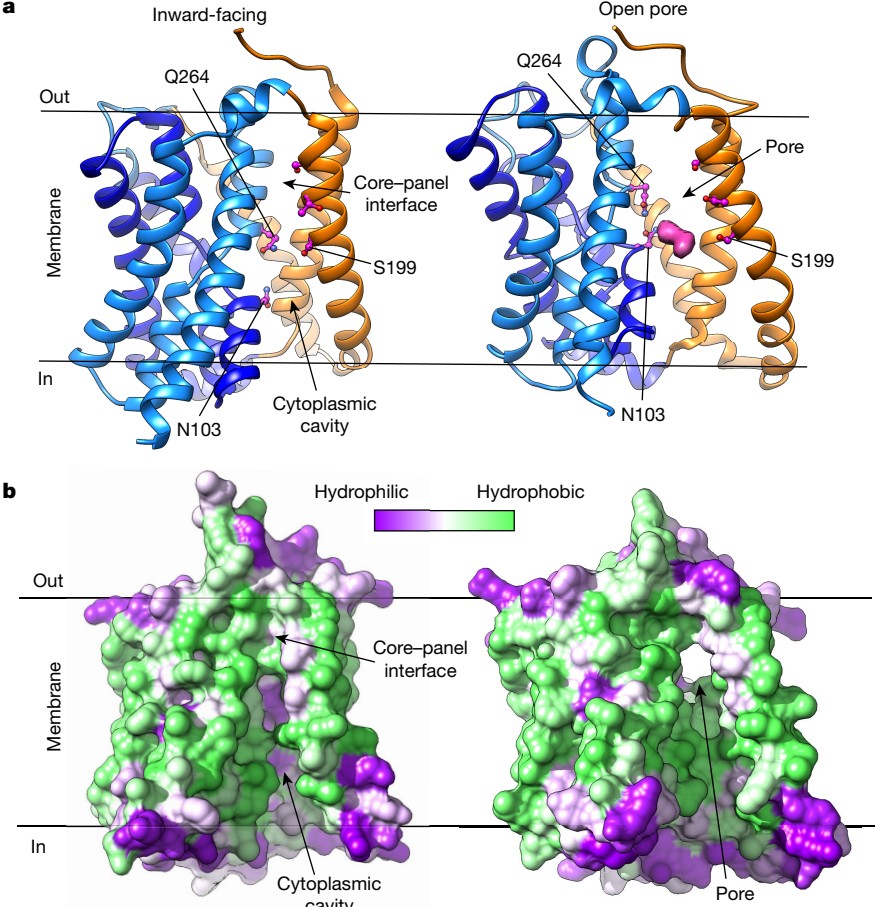

**Fig. 3 | Isomerization between open-pore and inward-facing states.**
**a**, NTCP$_{EM}$ structures in complex with Nb87 (left) and Mb91 (right) adopting inward-facing and open-pore conformations, respectively. Nb87 and Mb91 are not shown. Additional cryo-EM density in the open-pore structure is shown (pink surface) and polar residues in close proximity to the density are labelled in both structures for comparison. **b**, Molecular surface representation of

NTCP$_{EM}$ inward-facing (left) and open-pore (right) structures. Surfaces are coloured on the basis of a residue hydrophobicity scale, with green representing the most hydrophobic residues and purple representing the most hydrophilic residues. The structures have been tilted by around 20° relative to those in **a** to better display the cytoplasmic cavity (left), and the open pore (right).

Most of the reported residues important for binding of sodium and substrate map to the core domain. Sodium-binding sites 1 (Na1; including S105, N106, T123 and E257 sidechains) and 2 (Na2; including Q68 and Q261), which were first observed in crystallographic studies of prokaryotic SLC10 homologues[27], are structurally conserved in NTCP$_{EM}$ (Fig. 2c). Structural conservation and NTCP mutagenesis[31,32] strongly suggest that the two sodium ions that are thermodynamically coupled to bile salt transport[33,34] bind to these sites. Beyond Na1 and Na2, mutations at residues in the X motif[27] (equivalent to N262) or in close proximity[32] (Q293) impaired transport function, suggesting a role in substrate binding. Consistently, the NTCP-inactivating mutation[35] S267F, which is associated with hypercholanaemia and vitamin D deficiency in humans[36] lays just above the X motif, and A64T[37] is close to the sodium-binding sites.

## NTCP inward-facing state

In complex with Nb87, NTCP$_{EM}$ adopts an inward-facing state with core and panel domains tightly packing against each other on the extracellular side of the membrane (Fig. 3a, b). On the intracellular side, the domains separate, uncovering an amphiphilic large cavity (molecular surface volume > 1,500 Å$^3$) that opens to the cytoplasm, as well as laterally to the hydrophobic core of the membrane through a crevice between TM6 and TM9. On the other side of the transporter,

TM1 and TM5 pack against the core domain, occluding the cavity from the membrane.

Na1 and Na2 face this cavity and localize behind the conserved X motif. In addition, the cavity is lined by several conserved polar residues from the core domain (including N103, N262, Q264 and Q293), some of which have been shown to be important for transport; hydrophobic residues are mostly located in the panel domain. Amino acid conservation, mutagenesis studies and the large volume of the cavity suggest that it is part of the substrate pathway on the cytoplasmic side. Consistently, a molecule of taurocholate has been reported to bind to the equivalent region in the structure of a prokaryotic homologue[27].

## Transition to the open-pore conformation

In complex with Mb91, NTCP$_{EM}$ shows a marked conformational change compared with the inward-facing state (Fig. 3a, b, Supplementary Video 1). Core and panel domains rotate around 20° and move approximately 5 Å towards opposite sides of the membrane as nearly rigid bodies. These movements are facilitated by conserved glycine and proline residues that act as hinges in the connecting loops, as well as in the ICH and ECH (Extended Data Fig. 5). As a consequence, the two domains separate from each other on both extracellular and cytoplasmic sides, and open a wide pore through the transporter, exposing Na$^+$-binding sites and X motif residues simultaneously to opposite

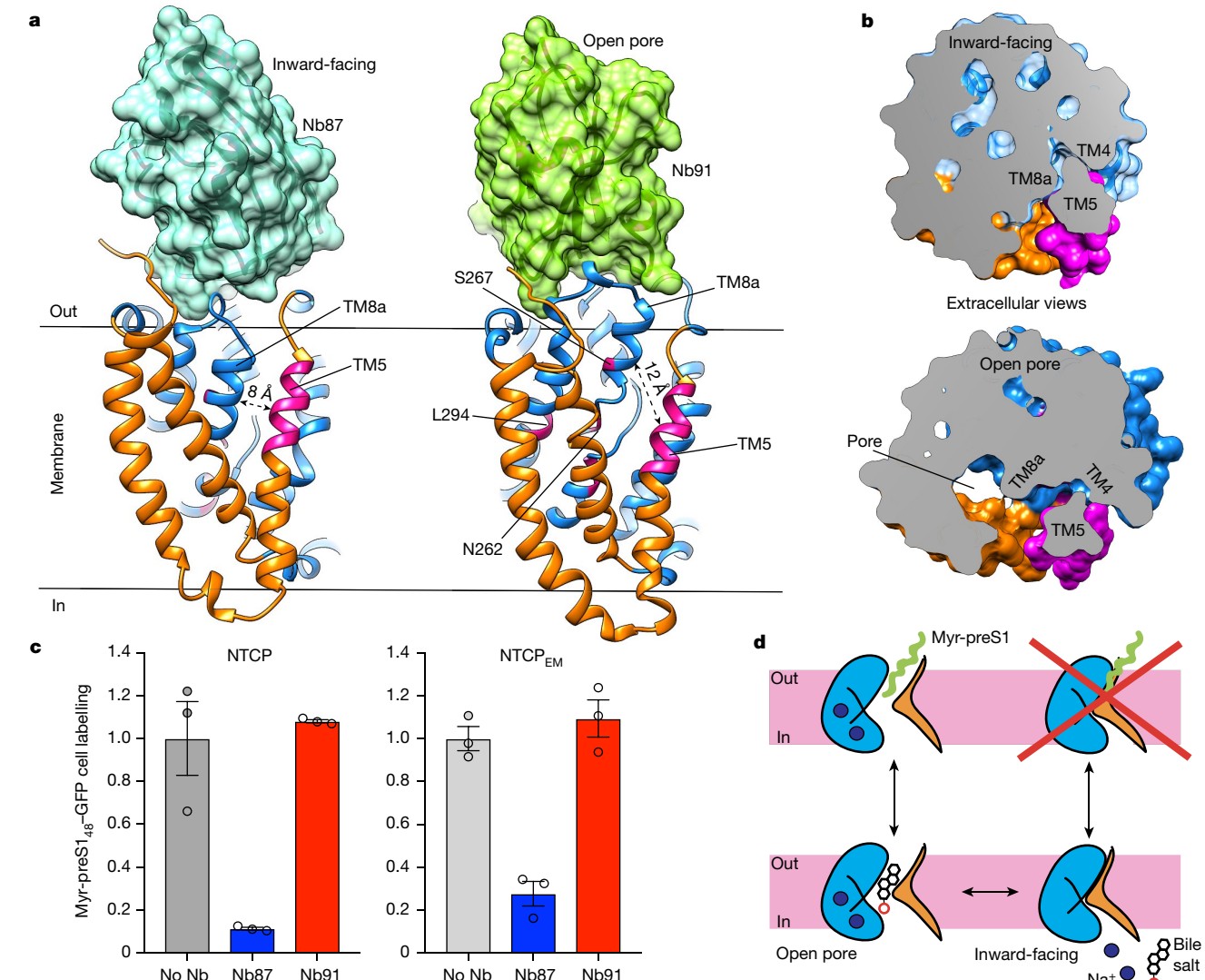

**Fig. 4 | Nb87 inhibits myr-preS1 binding. a**, Nb87 (left, cyan surface) and Nb91 (right, green surface) bind overlapping 3D epitopes on the extracellular surface of the core domain, distant from the myr-preS1 binding-determinant region in TM5 and residues within the pore (highlighted in pink). In the inward-facing state (left), TM5 is packed against the core domain (blue). In the open-pore state (right), the core domain moves outward and away from TM5, exposing important residues for myr-preS1 binding (pink). **b**, An extracellular view of cross-sections passing through the myr-preS1 binding-determinant region in TM5 (highlighted in pink). Inward-facing (top) and open-pore (bottom)

structures show the TM5–core domain (blue) interfaces. **c**, Myr-preS1$_{48}$–GFP labelling of cells expressing wild-type NTCP (left) and NTCP$_{EM}$ (right), respectively. Pre-incubation with Nb87, but not with Nb91, impaired myr-preS1$_{48}$–GFP labelling. Plots depict the mean of three biologically independent experiments and circles show values from individual experiments. Error bars represent s.e.m. **d**, Cartoon representation of the NTCP gated-pore transport mechanism and the relative movements of the core (blue) and panel (orange) domains. The myr-preS1 domain of HBV/HDV (green) preferentially binds to open-to-outside states of the NTCP transport cycle.

sides of the membrane. This is an unexpected conformational transition, as active transporters typically alternate exposure of their ligand binding sites to the extracellular and intracellular milieus, and adopt intermediate states with substrates occluded within the protein[38,39]. We discuss a plausible NTCP transport mechanism including an open-pore state below.

The surface lining the pore is amphiphilic, and most polar residues in this surface come from the core domain, including conserved side-chains in the X motif. Human NTCP mutations S267F and S199R—both of which are associated with hypercholanaemia[35,36,40]—also map to this surface, on opposite sides of the membrane. The pore has a minimum diameter of approximately 5 Å, and contains a large volume (2,400 Å$^3$), with its long axis oriented at an angle of about 45° to the membrane plane. It displays wide openings on both extracellular and intracellular sides to bulk solutions, as well as hydrophobic membrane leaflets. The amino acid conservation, the architecture and the amphiphilic

nature of the pore strongly suggest that it is the pathway for translocation of a wide range of amphiphilic bulky substrates transported by NTCP, including bile salts[7,41], sulfated steroids[42,43] and statins[14,15]. Consistently, in the cryo-EM map of NTCP$_{EM}$–Mb91, we observed extra density that partially occupies the pore on the cytoplasmic side, wedged in the crevice between TM6 and TM9 and in close proximity to conserved residues in the X motifs (Fig. 3a, Extended Data Fig. 7). Our cryo-EM sample included both Na$^+$ and substrate taurocholate, and the density probably corresponds to a substrate molecule bound to NTCP$_{EM}$. However, the lack of molecular features in the density precluded unambiguous determination of the bound molecule.

It is worth noting that the NTCP$_{EM}$–Mb91 complex structure was determined from samples in detergent solutions, raising the possibility that detergent molecules could have facilitated the open-pore state. To shed light on this question, we determined the cryo-EM structure of NTCP$_{EM}$–Nb91 complex reconstituted in nanodiscs. Despite the limited

resolution of the cryo-EM map (approximately 4.3 Å), we were able to confidently model NTCP$_{EM}$ in a conformation nearly identical to that observed in detergent solutions (r.m.s.d. ≈ 1.4 Å) (Extended Data Fig. 8), demonstrating that NTCP$_{EM}$ adopts an open-pore state in a lipid bilayer, which therefore represents a functional state of the transport cycle. We also observed similar extra density localized to the pore in the nanodisc-reconstituted NTCP$_{EM}$–Nb91 complex (Extended Data Fig. 7), further supporting the idea that the density corresponds to a substrate molecule, rather than detergent bound to the transporter.

## Nb87 impairs myr-preS1 binding

The conformational changes associated with NTCP$_{EM}$ pore opening have implications for the HBV/HDV receptor-recognition mechanism. A reported critical region for myr-preS1 binding and viral infection[2] (NTCP residues K157–L165) maps to the extracellular half of TM5 in the panel domain, and localizes far (more than 20 Å) from both Nb87- and Nb91- binding interfaces on the surface of the core domain (Fig. 4a). Notably, there is a conformational change around TM5 when comparing NTCP$_{EM}$ inward-facing and open-pore states (Fig. 4a, b, Supplementary Video 2). In the inward-facing state, TM5 packs tightly against TM4 and TM8b in the core, generating a shallow groove at the interdomain interface lined by hydrophobic residues. By contrast, relative movements of core and panel domains and tilting of TM5 towards the membrane in the open-pore state unpack the extracellular part of this helix away from the core domain (by as much as 4 Å), creating a crevice between TM5 on one side, and TM8b and the X motif on the other. Moreover, pore-lining residues that impair both myr-preS1 binding and bile salt transport[32] (including N262, S267 and L294) are accessible to the outside only in the open-pore state. The changes in accessibility around critical regions for HBV/HDV recognition suggests that myr-preS1 may bind differentially to open-pore and inward-facing states.

To test this hypothesis, we optimized a fluorescence-based myr-preS1 binding assay in cells using a purified myr-preS1$_{48}$ lipopeptide fused to GFP (myr-preS1$_{48}$–GFP). Indeed, myr-preS1$_{48}$–GFP labelling of cells expressing wild-type NTCP or NTCP$_{EM}$ was greatly decreased in the presence of Nb87, but was not affected by Nb91 (Fig. 4c). Nb87- and Nb91-overlapping epitopes on the surface of the core domain distant from HBV/HDV binding determinants strongly indicate that the inhibitory effect of Nb87 on myr-preS1$_{48}$–GFP binding is not owing to direct steric hindrance, but rather to stabilization of the inward-facing state that allosterically buries myr-preS1 binding determinants in the protein core. Overall, structural and functional results indicate that myr-preS1 binds preferentially to the open-pore state and interacts with exposed residues lining the pore at the interface between core and panel domains.

## Discussion

Our structural and functional analyses of NTCP$_{EM}$ in complexes with conformation-specific nanobodies reveal key molecular aspects of NTCP transport and HBV/HDV receptor-recognition mechanisms.

The NTCP$_{EM}$ open-pore structure is apparently at odds with the alternating-access transport mechanisms observed in most solute carrier families[38,39], including prokaryotic homologues of SLC10[27,28], which involve occluded substrate-bound intermediates of the transport cycle, raising the question of how to reconcile an open-pore intermediate state with thermodynamically active transport. Our structures suggest a plausible mechanism in which the pore is transiently open in the presence of substrate (and thermodynamically coupled Na$^+$) and closes upon release of ligands into the cytoplasm in the inward-facing state (Fig. 4d). The presence of an additional cryo-EM density in the pore, probably representing a bile salt molecule bound to the transporter, supports this type of mechanism. Moreover, sodium ions would contribute to gate the pore, avoiding bile salt permeation down its

electrochemical gradient and preferentially enabling bile salt binding at high extracellular sodium concentrations from the outside (under physiological ionic gradients), for instance by inducing outward-facing states that resemble those observed in prokaryotic homologues of SLC10[28]. Consistent with this line of thinking, early ion transport theories considered active carriers as pores whose gates are controlled by the energy source[44,45], challenging the classical distinction between channels and pumps[46]. To our knowledge, the NTCP open-pore state is the first structural demonstration of an active transporter displaying a wide-open-pore transport pathway for a bulky solute. Detailed knowledge of how the NTCP pore is gated on the extracellular side will require further structural and biophysical work.

The NTCP$_{EM}$ open-pore structure further shows that HBV/HDV-binding determinants line the pore within the membrane plane, accessible to the outside, and overlap with the substrate transport pathway. By sharp contrast, the inward-facing state shows tight packing of core and panel domains on the extracellular side burying virus-binding determinant residues within the protein core, and consistently, Nb87 antagonizes myr-preS1 binding. These results converge to suggest that myr-preS1 interacts with residues in the pore and, hence, that HBV/HDV selectively recognize NTCP conformations with an open-to-outside substrate pathway, while binding to inward-facing states is impaired (Fig. 4d). Such a mechanism explains the reported inhibitory effect of myr-preS1 binding on bile salt transport[32], as bound myr-preS1 would stabilize open-to-outside states and preclude isomerization to inward-facing ones and the antagonism between myr-preS1 and substrate binding[32], as both ligands would interact with overlapping binding sites within the pore.

The inhibitory effect of Nb87 on myr-preS1 binding reveals the therapeutic potential of molecules that stabilize NTCP inward-facing state(s), as allosteric inhibitors of viral cell entry. Such molecules could constitute alternative and/or synergistic therapeutic tools to existing lipopeptides that mimic high-affinity myr-preS1 binding[23,47], as well as neutralizing antibodies against HBV[48,49].

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

# Methods

## Thermostable NTCP constructs

Consensus amino acids were calculated using JALVIEW[50] and reported criteria[29] from sequences of representative NTCP vertebrate orthologues (Extended Data Fig. 1), aligned using Muscle[51]. Consensus amino acid exchanges were simultaneously introduced into wild-type NTCP sequence background with *N*-glycosylation mutations N5T and N11T, improving protein stability. Deletions of N-terminal residue E2, and the unstructured C terminus (residues T329–A349) in the consensus non-glycosylated construct further improved homogeneity of the sample, yielding the so-called NTCP$_{CO}$.

In general, the consensus approach generates protein samples with overall improved stability, but it is expected that by simultaneously introducing all consensus mutations, some destabilizing exchanges are included. To minimize the latter, we probed thermal stability of single-point NTCP$_{CO}$ mutants, in which we reverted consensus amino acids to the wild-type residues, using fluorescence-detection size-exclusion chromatography[52] (SEC). Removal of destabilizing consensus exchanges in NTCP$_{CO}$, yielded a consensus design, NTCP$_{EM}$, which is nearly identical to wild-type NTCP (approximately 98% identity) (Extended Data Figs. 1, 6), while preserving Na$^+$-dependent bile salt transport as well as myr-preS1 recognition mechanisms.

## Protein expression and purification

cDNAs encoding NTCP constructs were synthesized (GenScript) and subcloned into a pcDNA3.1(+) vector encompassing a C-terminal PreScission site, followed by GFP, and two Strep-tags in tandem for affinity purification. Protein expression was done in HEK 293F cells (Thermo Fisher; cells were not authenticated or tested for mycoplasma contamination) by transient transfection, as described[53] with small variations. In brief, cells grown in FreeStyle 293 medium (Thermo Scientific) were transfected with linear 25K polyethyleneimine (PEI) (Polysciences) at a cell density of $2.5 \times 10^6$ cells per ml using 3 µg ml$^{-1}$ DNA. Valproic acid (VPA) was added to the culture at a final concentration of 2.2 mM 6–12 h after transfection and cells were grown for additional 48 h before collection.

Cell pellets were resuspended and lysed in buffer containing 50 mM HEPES pH 7.4, 200 mM NaCl, 5% v/v glycerol, 1 mM EDTA, 1 mM TCEP, 0.5 mM sodium taurocholate and supplemented with protease inhibitors (1 mM PMSF and protease inhibitor cocktail from Sigma), 1% dodecyl-β-D-maltopyranoside (DDM) (Anatrace) and 0.2% cholesteryl hemi-succinate tris salt (CHS) (Anatrace), and incubated for 1 h. Cell debris was removed by ultracentrifugation. Detergent-solubilized transporters were purified by affinity chromatography using streptactin sepharose resin (Cytiva Life Sciences). Resin was pre-equilibrated in buffer A containing 50 mM HEPES pH 7.4, 200 mM NaCl, 5% v/v glycerol, 0.017% DDM, 0.0034% CHS, and 0.2 mM sodium taurocholate, and incubated with transporters for 1 h under rotation. Resin was extensively washed with buffer A, and protein was eluted in buffer B containing 50 mM HEPES pH 7.4, 200 mM NaCl, 5% v/v glycerol, 0.017% DDM, 0.0034% CHS, 0.2 mM sodium taurocholate, and 2.5 mM desthiobiotin. The eluted protein was digested with PreScission protease overnight, concentrated to several mg per ml using 100 kDa MWCO concentrator (Corning Spin-X UF concentrators) and injected in a Superose 6 column (GE Healthcare Life Sciences) using SEC buffer containing 20 mM HEPES pH 7.4, 100 mM NaCl, 0.017% DDM, 0.0034% CHS, and 0.2 mM sodium taurocholate. Purified transporters were used immediately or flash frozen and stored at −80 °C. All purification steps were done at 4 °C.

NTCP$_{EM}$ complexes with nanobodies and megabodies, respectively, were formed by mixing purified protein samples at 1:1.2 (transporter:nanobody, or megabody) molar ratio, and incubated for 2h at 4 °C. Excess nanobody or megabody was removed by SEC using SEC buffer. MSP1D1 nanodisc-scaffold protein was expressed and purified using published protocols[54]. Reconstitution was done by mixing purified NTCP$_{EM}$–Nb and NTCP$_{EM}$–Mb complexes, respectively, with MSP1D1 and liver total lipid extract (Avanti Polar Lipids) at 0.1:1:15 molar ratio, and incubated with methanol-activated biobeads for 2 h. Biobeads were exchanged once, and the mixture was further incubated overnight. Nanodisc-reconstituted sample was purified in a Superdex 200 increase column (GE Healthcare Life Sciences) in buffer containing 20 mM HEPES pH 7.4, 100 mM NaCl, and 0.2 mM sodium taurocholate. Samples were concentrated as described above, and immediately used for cryo-EM grid preparation.

## Nanobody generation, expression and purification

Nanobodies against NTCP$_{CO}$ were generated using published protocols[55]. In brief, one llama (*Lama glama*) was six times immunized with a total 0.9 mg of NTCP$_{CO}$ reconstituted in proteoliposomes. Four days after the final boost, blood was taken from the llama to isolate peripheral blood lymphocytes. RNA was purified from these lymphocytes and reverse transcribed by PCR to obtain the cDNA of the open reading frames coding for the nanobodies. The resulting library was cloned into the phage display vector pMESy4 bearing a C-terminal His$_6$ tag and a CaptureSelect sequence tag (Glu-Pro-Glu-Ala). Different nanobody families, as defined by the difference in the CDR3, were selected by biopanning. For this, NTCP$_{CO}$ reconstituted in proteoliposomes was solid phase coated directly on plates. NTCP$_{CO}$ specific phage were recovered by limited trypsinization, and after two rounds of selection, periplasmic extracts were made and analysed using ELISA screens. Nb87 and Nb91 were expressed in *Escherichia coli* for subsequent purification from the bacterial periplasm. After Ni-NTA (Sigma) affinity purification, nanobodies were further purified by SEC in buffer: 10 mM HEPES pH 7.4, and 110 mM NaCl.

Nb91 was enlarged by fusion to the circular permutated glucosidase of *E. coli* K12 (YgjK, 86 kDa) to build the megabody referred to as Mb91. Mb91 was generated and purified using previously described protocols[30].

## Fluorescent substrate analogue transport assay

Sodium-dependent substrate uptake was measured in HEK 293F cells transfected with 2 µg µl$^{-1}$ cDNA using the above-mentioned protocol with small modifications. Forty-eight h after transfection, around 1 million cells were pelleted, washed, and resuspended in 500 µl of transport buffer (110 mM NaCl, 4 mM KCl, 1 mM MgSO$_4$, 1 mM CaCl$_2$, 45 mM mannitol, 5 mM glucose and 10 mM HEPES pH 7.4), or control buffer in which NaCl was substituted with choline chloride (ChCl). To probe the effect of nanobodies on bile salt transport, cells were incubated with nanobodies for 1.5 h, followed by addition of the fluorescent substrate analogue tauro-nor-THCA-24-DBD[56,57] (tebu-bio) to a final concentration of 10 µM for 30 min at 37 °C. Excess fluorescent analogue was removed by centrifugation (13,000$g$ for 30 s), and 1 wash with the above-mentioned control buffer. Then, cells were resuspended and lysed using Pierce IP lysis buffer (Thermo Fisher). Finally, lysates were centrifuged (13,000$g$ for 10 min), and transferred to black 96-well flat-bottom plates (Grenier), and quantified by fluorescence in a micro-plate reader (CLARIOstar-Plus) using excitation at 454 nm and emission of 570 nm. Three biologically independent experiments were quantified in triplicate samples. Nb titrations data were fitted in Prism 8.0.1 (GraphPad) to the following dose-respond curve:

$$\text{Fractional transport} = 1 + \frac{Y_{\min} - 1}{1 + 10^{\log IC_{50} - x}}$$

Where $Y_{\min}$ corresponds to fraction of transport at saturating Nb concentrations, IC$_{50}$ is the half-maximal inhibitory concentration, and $x$ is log[Nb].

## Myr-preS1 purification and binding assay

cDNA encoding the N-terminal myristoylated consensus residues 2–48 of human HBV myr-preS1 domain (myr-GTNLSVPNPLGFFPDHQL DPAFRANSNNPDWDFNPNKDHWPEANKVG) was synthesized (GenScript)

and subcloned into a pcDNA3.1(+) vector encompassing a C-terminal GFP, and poly Histidine-tag (namely, myr-preS1$_{48}$–GFP). Myr-preS1$_{48}$–GFP was expressed in HEK 293F cells (Thermo Fisher) by transient transfection, as described for expression of NTCP$_{EM}$ purification. Cells were lysed by 3–5 passes through a homogenizer (EmulsiFlex-C5, Avestin) and membrane fraction was collected by ultracentrifugation. Membranes were resuspended in a buffer containing 50 mM HEPES pH 7.4, 200 mM NaCl, 5% v/v glycerol, 1 mM EDTA, protease inhibitors (1 mM PMSF and protease-inhibitor cocktail from Sigma), 1% dodecyl-β-ᴅ-maltopyranoside (Anatrace), and 0.2% cholesteryl hemi-succinate tris salt (Anatrace), and incubated for 1 h. Solubilized myr-preS1$_{48}$–GFP was subjected to ultracentrifugation and then purified by affinity chromatography using anti-His affinity resin (Sigma). Resin was pre-equilibrated in a buffer containing 50 mM HEPES pH 7.4, 200 mM NaCl, 5% v/v glycerol, 0.013% DDM, 0.0027% CHS, and incubated with detergent-solubilized myr-preS1$_{48}$–GFP for 1 h under rotation. Resin was extensively washed with buffer containing 50 mM HEPES pH 7.4, 200 mM NaCl, 5% v/v glycerol, 0.013% DDM, 0.0027% CHS and 50 mM imidazole. Myr-preS1$_{48}$–GFP was eluted in buffer containing 50 mM HEPES pH 7.4, 200 mM NaCl, 5% v/v glycerol, 0.013% DDM, 0.0027% CHS, and 250 mM imidazole. The eluted protein was concentrated to several mg ml$^{-1}$ using a 30 kDa MWCO concentrator (Corning Spin-X UF concentrators) and injected into a Superose 6 column (GE Healthcare Life Sciences) using a SEC buffer containing 20 mM HEPES pH 7.4, 200 mM NaCl, 0.013% DDM, and 0.0027% CHS. Myristoylation of the sample was confirmed by mass spectrometry. Purified myr-preS1$_{48}$–GFP was flash frozen and stored at −80 °C. All purification steps were done at 4 °C.

Myr-preS1$_{48}$–GFP binding to NTCP constructs was assayed in HEK 293F cells, grown and transfected with 1 µg ml$^{-1}$ DNA using the protocol described above. Forty-eight hours after transfection, cells were washed with pre-warmed PBS, and ~1 million cells were pelleted and resuspended in 1 ml of PBS. To probe the effect of nanobodies, cells expressing NTCP constructs were pre-incubated with 10 µM nanobodies for 1.5 h. They were then labelled with 10 nM (wild-type NTCP) or 50 nM (NTCP$_{EM}$) purified myr-preS1$_{48}$–GFP for 30 min. Excess fluorescent-probe was removed by centrifugation (13,000$g$ for 30 s), and one wash with PBS. Cells were then re-suspended in PBS and GFP fluorescence was recorded in a micro-plate reader (CLARIOstar-Plus) using excitation at 470 nm and emission at 508 nm.

## Electron microscopy sample preparation and data acquisition

Purified NTCP$_{EM}$–Nb or –Mb complexes were applied to glow-discharged Au 300 mesh Quantifoil R1.2/1.3. Typically, 4 µl of sample at 3-4 mg ml$^{-1}$ was applied to the grids, and the Vitrobot chamber was maintained at 100% humidity and 4 °C. Grids were screened in 200 kV Talos Arctica microscope (ThermoFisher) at the IECB cryo-EM imaging facility. Final data collection was performed in 300 kV Titan Krios microscope (ThermoFisher) at EMBL-Heidelberg Cryo-Electron Microscopy Service Platform, equipped with K3 direct electron detector (Gatan). Final images were recorded with SerialEM[58] at a pixel size of 0.504 Å. Dose rate was 15–20 e$^-$ pixel s$^{-1}$.

For NTCP$_{EM}$–Nb87 complexes, 21,792 images were recorded with −0.5 to −1.5 µm defocus range. Images were collected with 0.7-s subframes (total 40 subframes), corresponding to a total dose of 57.8 e$^-$ Å$^{-2}$. For NTCP$_{EM}$–Mb91 complexes, 21,390 images were recorded with −0.6 to −1.75-µm defocus range. Images were collected with 0.7 s subframes (total 40 subframes), corresponding to a total dose of 56.5 e$^-$ Å$^{-2}$.

## Cryo-EM data processing, model building and structure analysis

All datasets were processed with cryoSPARC v2 and v3[59]. Movies were gain corrected, and aligned using in-built patch-motion correction routine. Contrast transfer function (CTF) parameters were estimated using the in-built patch-CTF routine in cryoSPARC. Low-quality images were discarded manually upon visual inspection.

For the NTCP$_{EM}$–Mb91 complex, 5,796,802 particles were template-picked from 21,390 micrographs, and selected through several rounds of 2D, as well as 3D ab initio classifications. Particles from 3D ab initio classes displaying interpretable density for transmembrane helices were pooled, and used for homogenous refinement (Extended Data Fig. 2). Cryo-EM density corresponding to both detergent micelle and megabody scaffold were masked out, and particles were further subjected to local refinement using a fulcrum that localized to center of NTCP$_{EM}$ transmembrane region. Focused refinement yielded a final map at an overall resolution of 3.3 Å, based on the gold-standard 0.143 Fourier shell correlation (FSC) cut-off.

For the NTCP$_{EM}$–Nb87 complex, 6,535,687 particles were template-picked from 21,792 micrographs, and classified through several rounds of 2D and 3D ab initio classifications (Extended Data Fig. 3). Around 220,000 selected particles were further classified by heterogenous refinement, yielding a final set of 61,053 particles that were processed by non-uniform refinement[60]. Further focused refinement excluding nanodisc scaffold yielded a final map at an overall resolution of 3.7 Å, based on the gold-standard 0.143 FSC cut-off. Maps were visualized using UCSF Chimera[61] and ChimeraX[62].

The cryo-EM map of the NTCP$_{EM}$–Mb91 complex showed clear density for most sidechains in the transmembrane helices, although TM1 and TM6 in the panel domain displayed fewer molecular features, and was used to build an atomic model of NTCP$_{EM}$ using Coot[63,64]. Secondary structure predictions using Psipred[65] and bacterial homologue structure (Protein Data Bank ID 3ZUY) were used to help initial sequence assignment. Initial Nb models were created with I-TASSER[66], and then fit as rigid bodies into the density, followed by manual building and modification in Coot[63,64]. The inward-facing conformation in the NTCP$_{EM}$–Nb87 complex was built by fitting core and panel domains from NTCP$_{EM}$–Mb91 structure as separate rigid bodies into the density, followed by manual modification in Coot. All atomic models were refined using PHENIX[67].

Structural analyses were carried out as follows: protein cavity calculations with CASTp 3.0[68], pore calculations MOLEonline 2.5[69], protein–protein interfaces with PISA[70], and amino acid conservation surface mapping with ConSurf[71].

### Reporting summary

Further information on research design is available in the Nature Research Reporting Summary linked to this paper.

## Data availability

Structural models of NTCP$_{EM}$–Nb87 and NTCP$_{EM}$–Mb91 complexes have been deposited in the Protein Data Bank (PDB) with accession codes 7PQG and 7PQQ, respectively, and the corresponding cryo-EM maps were deposited in the Electron Microscopy Data Bank (EMDB) under accession numbers EMD-13593 and EMD-13596. Materials are available upon reasonable request and signing of a Material Transfer Agreement.

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

**Acknowledgements** We acknowledge the European Institute of Chemistry and Biology cryo-EM imaging facility for support with cryo-EM sample screening and initial data acquisition, the European Molecular Biology Heidelberg Cryo-Electron Microscopy Service Platform (for NTCP$_{EM}$–Mb91 and NTCP$_{EM}$–Nb87), and the Institut Pasteur cryo-EM Nanoimaging Facility (for NTCP$_{EM}$–Nb91) for support in image acquisition. Access to Institut Pasteur cryo-EM Nanoimaging Facility was partly supported by CACSICE infrastructure grant (ANR-11-EQPX-008). We thank K. Willebal and Nanobody4instruct center for technical support in nanobody generation and T. Uchanski for the Mb construction. This research was funded by the European Research Council (ERC) under European Union Horizon 2020 Program (grant 771965) with additional contributions from ANRS (grant 17125/18141), IDEX Senior Chair Universite de Bordeaux, and Region Nouvelle-Aquitaine (grant 8166910) (all to N.R.). Nanobody generation was funded by Instruct-Eric (PIDs: 2036, 15056, 7370 and 7111 to N.R.).

**Author contributions** generated nanobodies. K.G. and N.R. analysed functional data and structural models, and prepared the manuscript with comments from all authors. N.R. conceived and supervised the project.

**Competing interests** K.G., F.S.I., E.P., J.S. and N.R. are listed as co-inventors on a patent application (22151078.7) by Institut Pasteur and VIB-VUB Center for Structural Biology related to the nanobodies used in this work.

**Additional information**
**Correspondence and requests for materials** should be addressed to Nicolas Reyes.

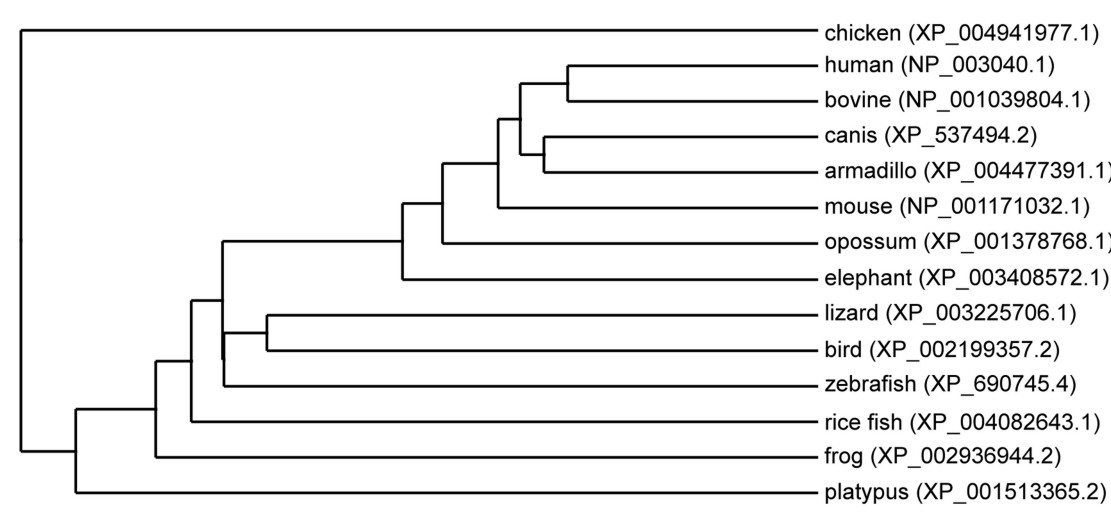

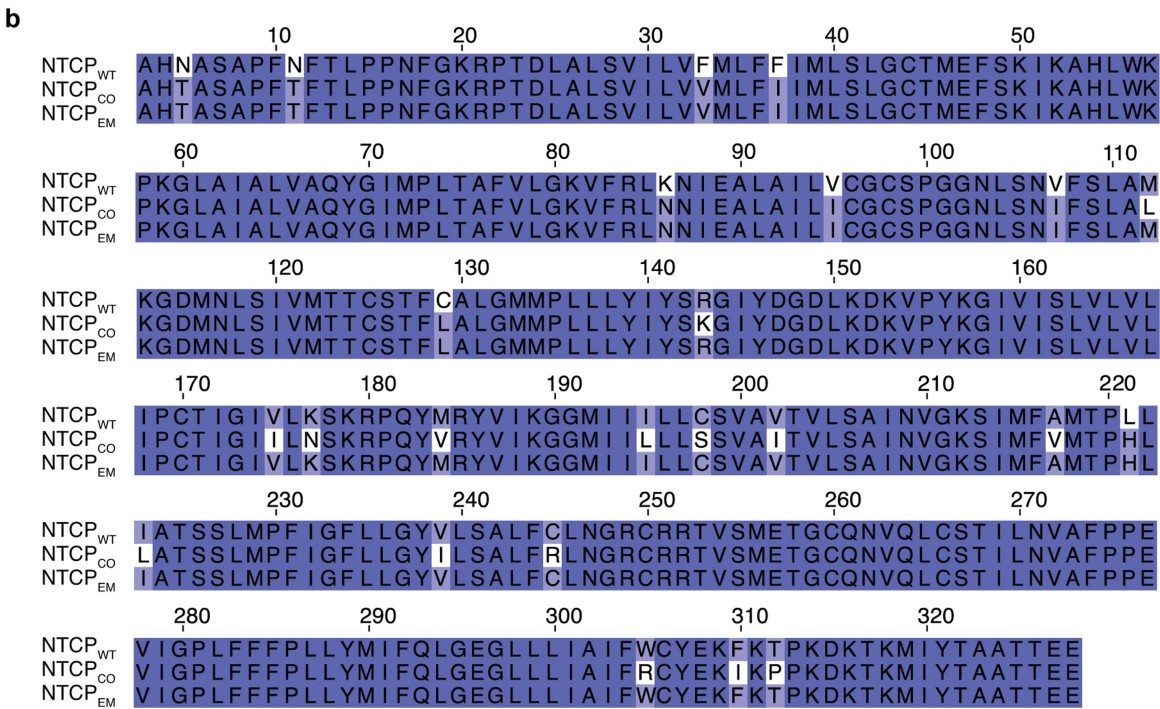

**Extended Data Fig. 1 | NTCP consensus designs. a**, Phylogenetic tree of NTCP vertebrate orthologs used to determine consensus amino acids. NCBI protein sequence IDs are given in parenthesis. **b**, Amino acid sequence alignment of human NTCP (residues 3–328) and consensus designs $NTCP_{CO}$ and $NTCP_{EM}$.

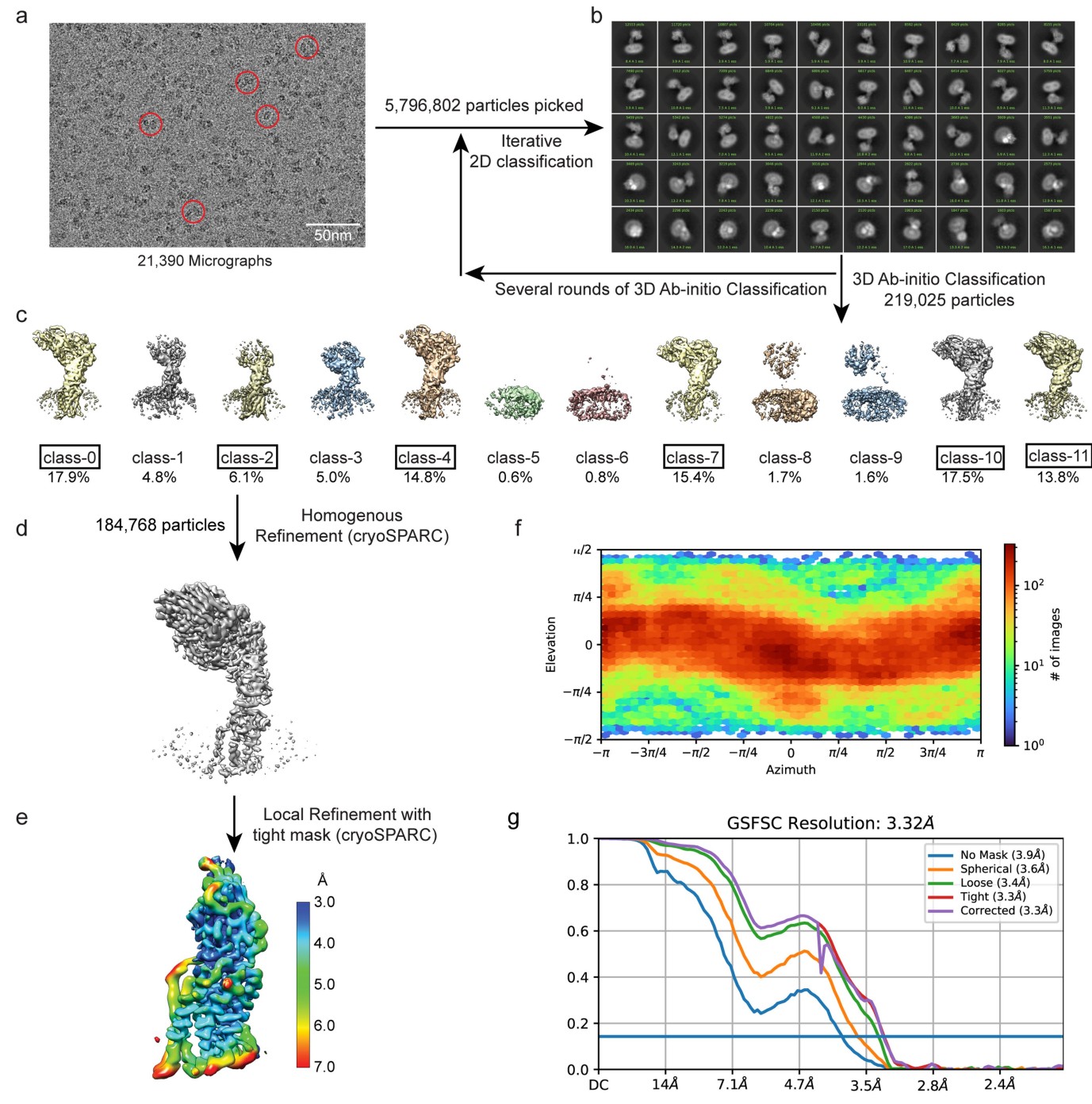

**Extended Data Fig. 2 | Cryo-EM data processing pipeline of NTCP_EM-Mb91 complex. a**, Representative EM micrograph with examples of individual particles (red circles). 21,390 micrographs were collected. **b**, Gallery of representative 2D class-averages. **c**, 3D classes from *ab initio* classification.

**d**, Homogenous refined map **e**, Local-refinement map after micelle removal, and color coded according to local resolution estimation. **f**, Viewing direction distribution plot. **g**, Fourier shell correlation (FSC) plot of local refinement with FSC threshold at 0.143.

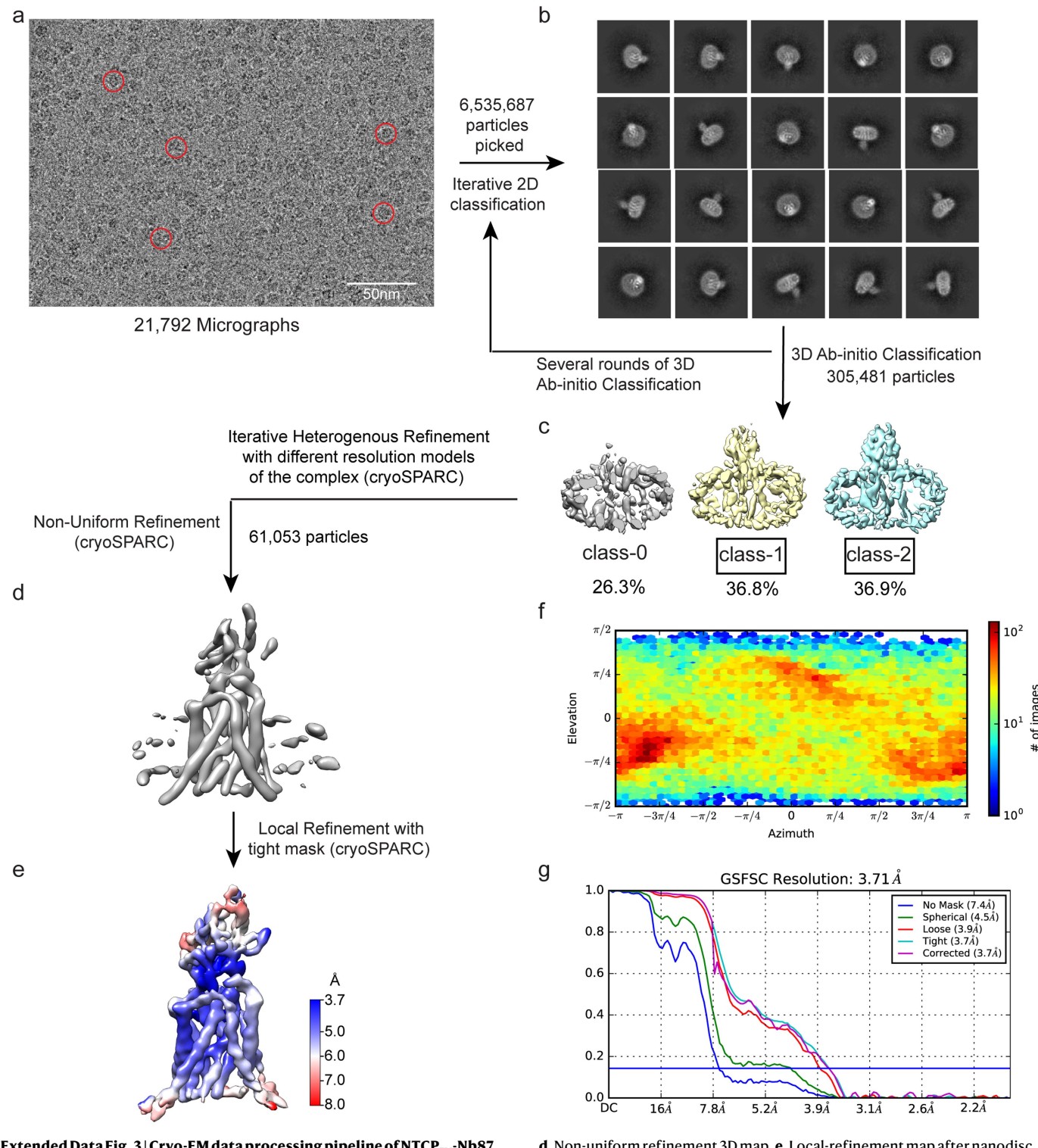

**Extended Data Fig. 3 | Cryo-EM data processing pipeline of NTCP_EM-Nb87 complex. a**, Representative EM micrograph with examples of individual particles (red circles). 21,792 micrographs were collected. **b**, Gallery of representative 2D class-averages. **c**, 3D classes from *ab initio* classification.

**d**, Non-uniform refinement 3D map. **e**, Local-refinement map after nanodisc removal, and color coded according to local resolution estimation. **f**, Viewing direction distribution plot. **g**, Fourier shell correlation (FSC) plot of local refinement with FSC threshold at 0.143.

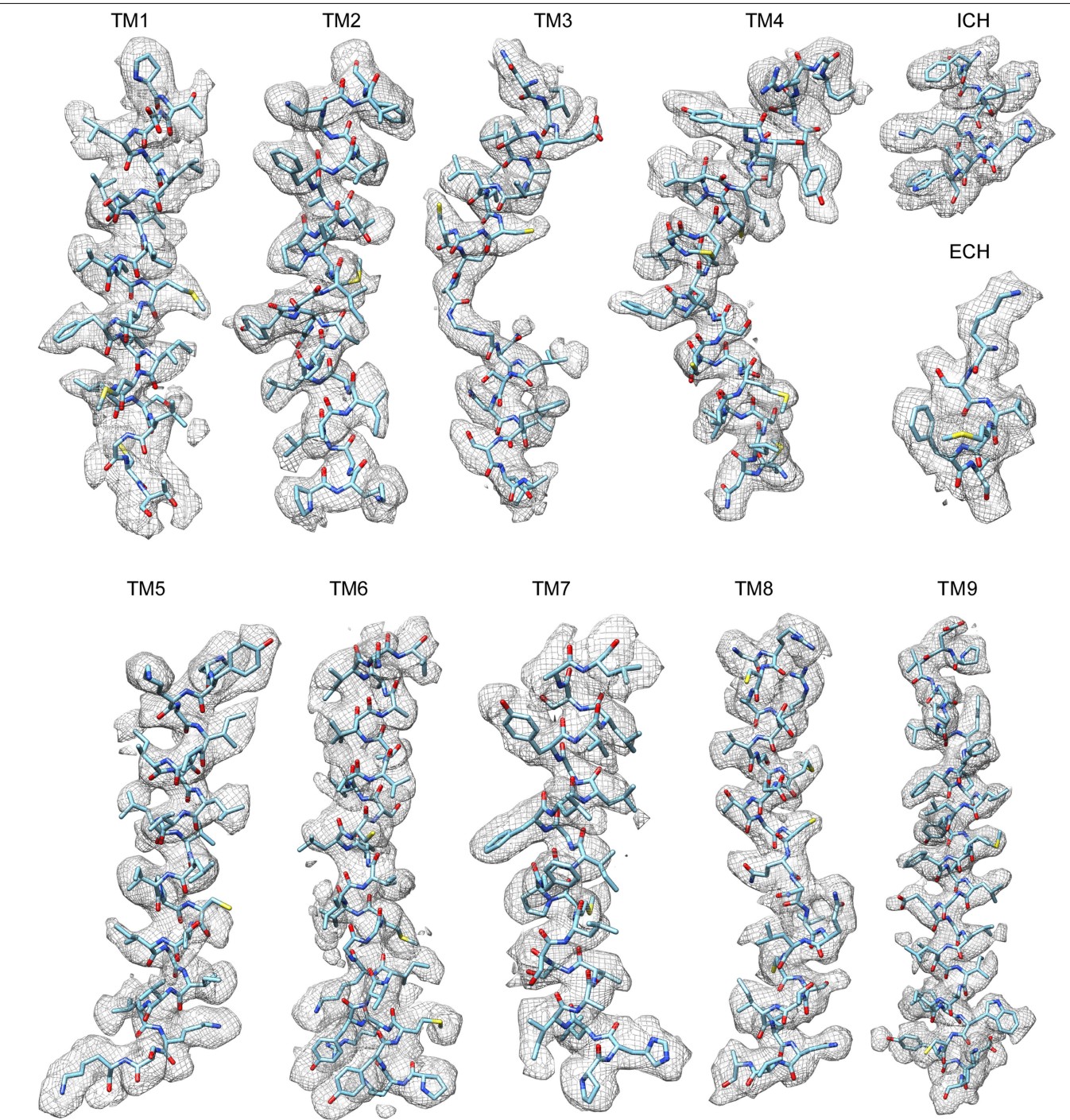

**Extended Data Fig. 4 | Cryo-EM density in NTCP_EM-Mb91 structure.** Cryo-EM density corresponding to individual NTCP_EM transmembrane helices in complex with Mb91.

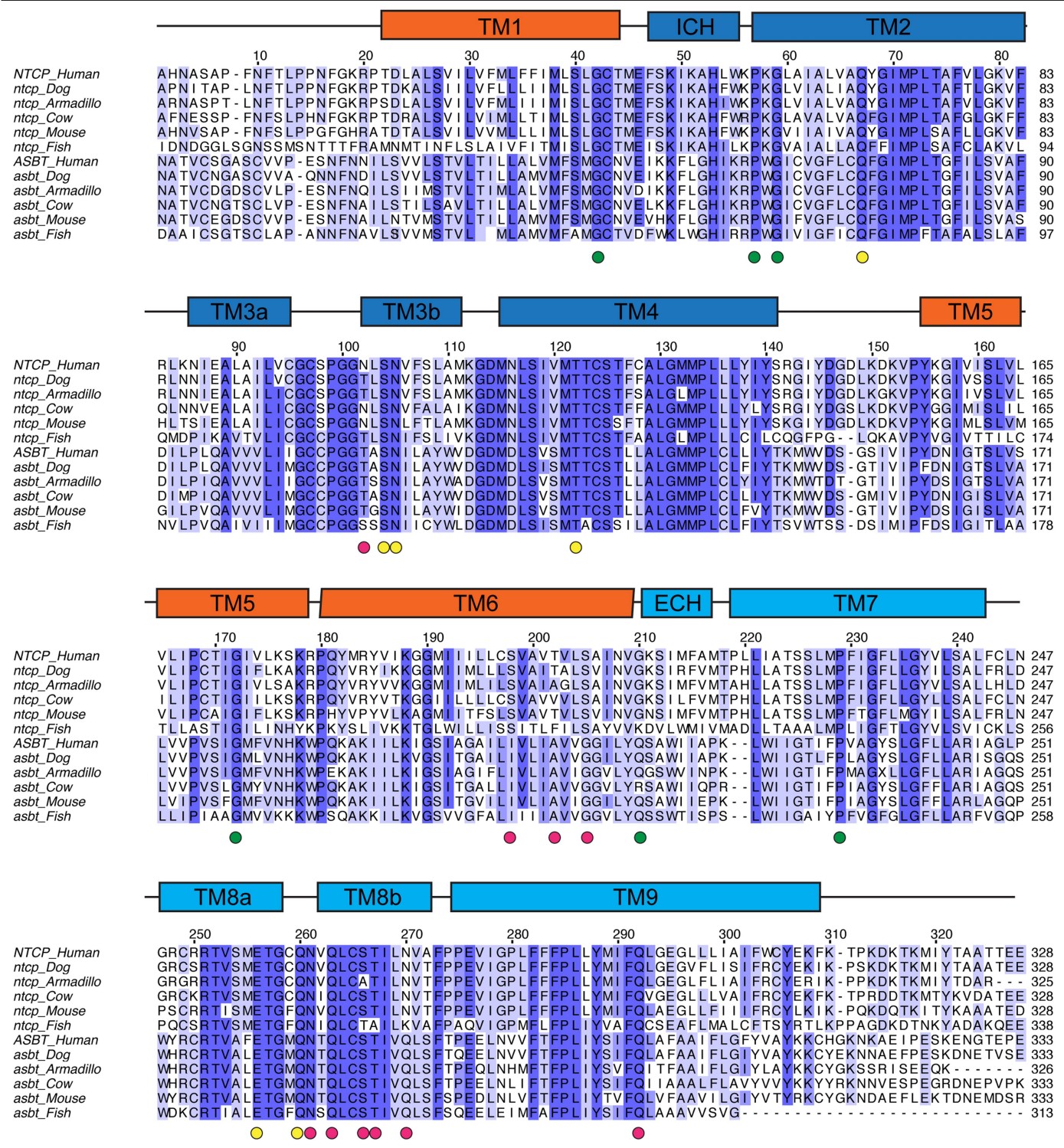

**Extended Data Fig. 5 | Amino acid sequence alignment of NTCP and ASBT vertebrate orthologs.** Yellow circles indicate residues that contribute sidechains to Na1 and Na2, pink circles conserved polar residues in NTCP orthologs that line the space between core and panel domains, and green circles proline and glycine residues that act as hinges during the isomerization between open-pore and inward-facing states.

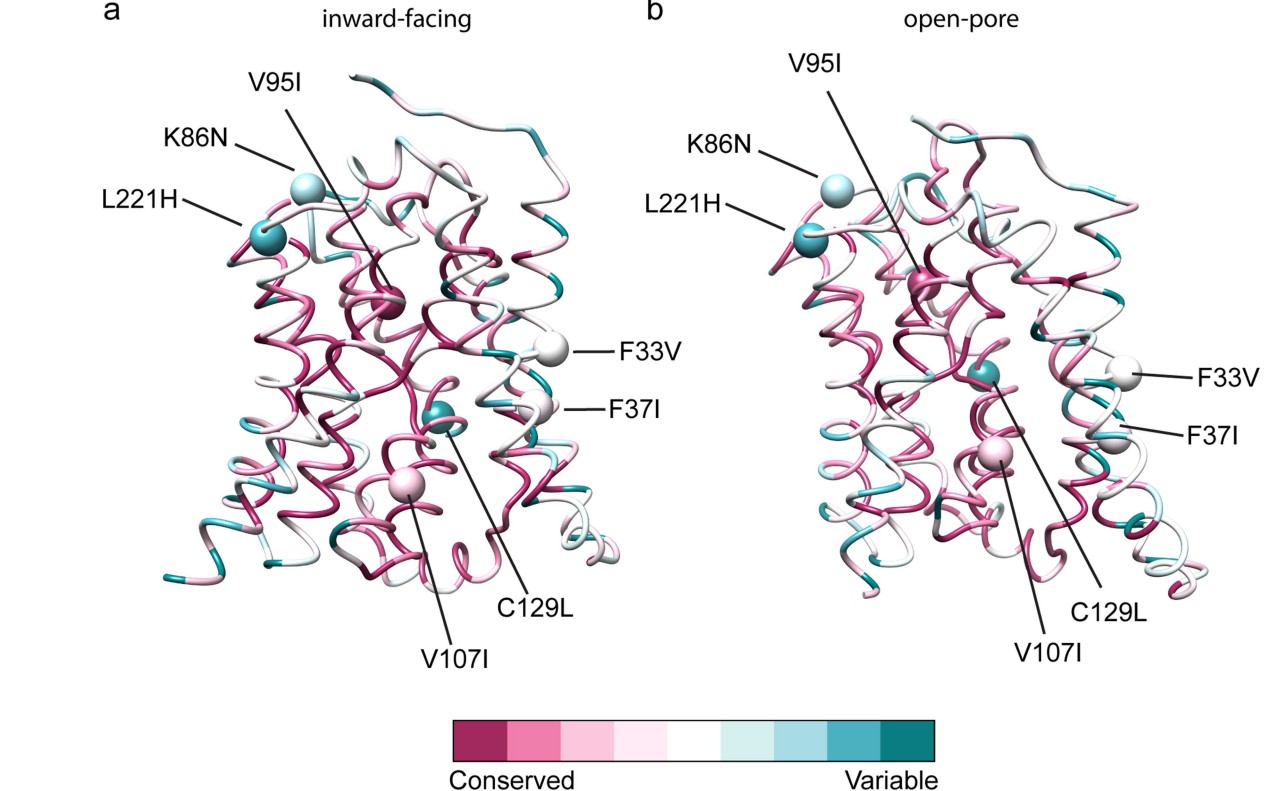

a   inward-facing

V95I
K86N
L221H
F33V
F37I
C129L
V107I

b   open-pore

V95I
K86N
L221H
F33V
F37I
C129L
V107I

Conserved                    Variable

**Extended Data Fig. 6 | NTCP amino acid conservation surface mapping.**
Amino acid conservation across NTCP vertebrate orthologs is mapped into
NTCP$_{EM}$ inward-facing (left) and open-pore (right) structures. Spheres
represent alpha-carbon atoms of 7 consensus mutations to improve thermal
stability in the construct used for cryo-EM analysis (NTCP$_{EM}$).

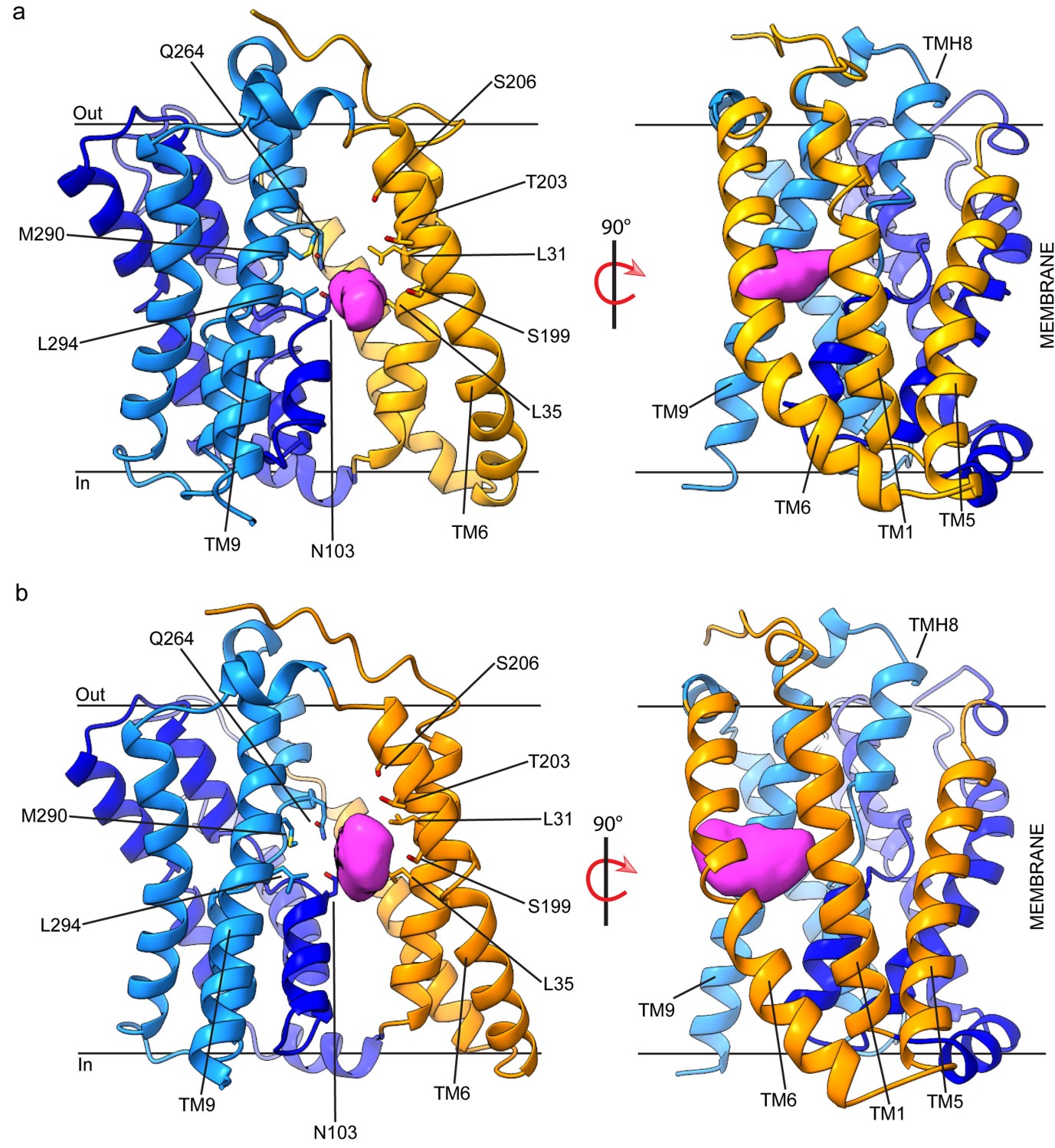

**Extended Data Fig. 7 | Extra-density in NTCP$_{EM}$ open-pore structures.** **a**, Membrane views of NTCP$_{EM}$-Mb91 in detergent solutions highlighting extra cryo-EM density in the pore (purple surface), and residues in proximity of the density. **b**, Membrane views of NTCP$_{EM}$-Nb91 in nanodisc highlighting similar extra cryo-EM density in the pore, and residues in proximity.

a

b

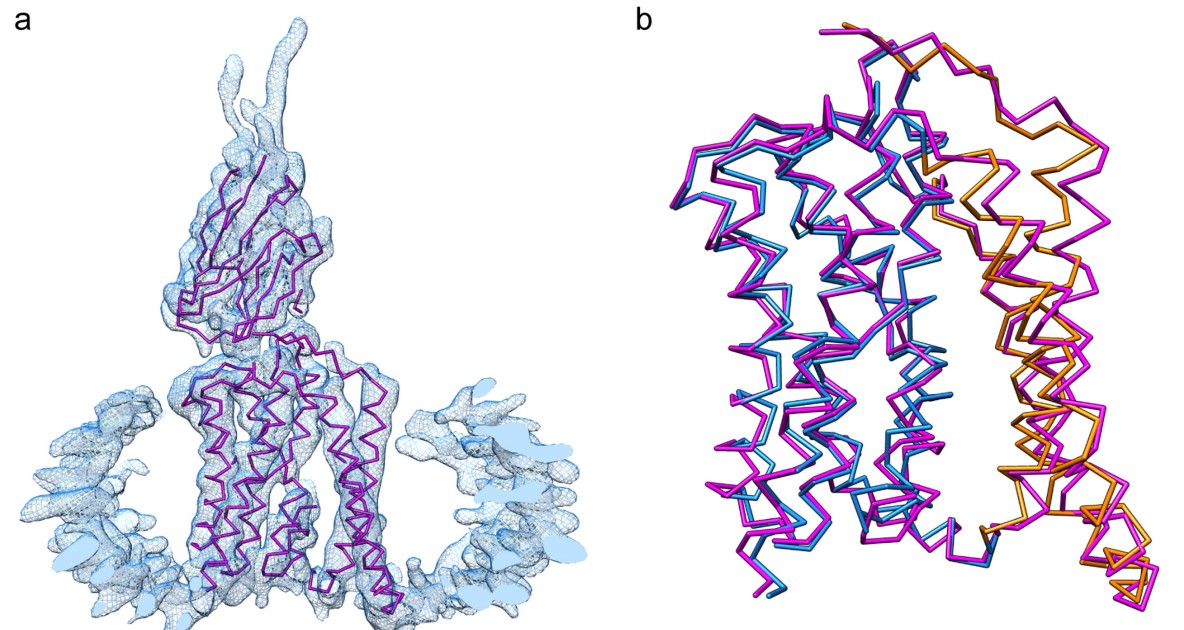

**Extended Data Fig. 8 | Cryo-EM density and structure of NTCP$_{EM}$-Nb91 reconstituted in nanodisc. a**, Structure of NTCP$_{EM}$-Nb91 complex reconstituted in nanodisc and the corresponding cryo-EM density.

**b**, Superimposition of NTCP$_{EM}$ structures in complex with Nb91 in nanodisc (dark pink) and that in complex with Mb91 in detergent (core domain, blue; panel domain, orange) (RMSD ~1.4 Å over all atoms).

**Extended Data Table 1 | Cryo-EM data collection, refinement and validation statistics**

|  | #1 NTCP$_{EM}$-Mb91 (EMDB-13596) (PDB 7PQQ) | #2 NTCP$_{EM}$-Nb87 (EMDB-13593) (PDB 7PQG) |
|---|---|---|
| **Sample preparation, Data collection and processing** | | |
| Medium for sample preparation | DDM | Nanodisc |
| Microscope | Titan Krios 2 | Titan Krios 2 |
| TEM mode | EFTEM Nanoprobe | EFTEM Nanoprobe |
| Imaging Mode | Counting | Counting |
| Camera | Gatan Quantum-K3 | K3 |
| Magnification | 165,000x | 165,000x |
| Voltage (kV) | 300 | 300 |
| Electron exposure (e–/Å$^2$) | 56.5 | 57.8 |
| Defocus range (µm) | -0.6 to -1.75 | -0.5 to -1.5 |
| Pixel size (Å) | 0.504 | 0.504 |
| Symmetry imposed | C1 | C1 |
| Initial particle images (no.) | 5,796,802 | 6,535,687 |
| Final particle images (no.) | 184,768 | 61,053 |
| Map resolution (Å) | 3.3 | 3.7 |
| FSC threshold | 0.143 | 0.143 |
| Map resolution range (Å) | 22.3-2.8 | 19.9-3.4 |
| **Refinement** | | |
| Initial model used (PDB code) | none | 7PQQ |
| Model resolution (Å) | 3.2 | 3.7 |
| FSC threshold | 0.143 | 0.143 |
| Model resolution range (Å) | 22.3-2.8 | 19.9-3.4 |
| Map sharpening $B$ factor (Å$^2$) | -60.3 | -18.1 |
| Model composition | | |
| Non-hydrogen atoms | 3,207 | 3,251 |
| Ligands | none | none |
| $B$ factors (Å$^2$) | | |
| Protein | 43.57 | 156.22 |
| Ligand | - | - |
| R.m.s. deviations | | |
| Bond lengths (Å) | 0.004 (0) | 0.004 (0) |
| Bond angles (°) | 0.678 (0) | 0.877 (0) |
| Validation | | |
| MolProbity score | 1.92 | 2.39 |
| Clashscore | 12.2 | 27.61 |
| Poor rotamers (%) | 0 | 0.28 |
| Ramachandran plot | | |
| Favored (%) | 95.38 | 92.6 |
| Allowed (%) | 4.62 | 7.40 |
| Disallowed (%) | 0 | 0 |

# Reporting Summary

## Statistics

For all statistical analyses, confirm that the following items are present in the figure legend, table legend, main text, or Methods section.

| n/a | Confirmed | |
|---|---|---|
| ☐ | ☒ | The exact sample size (*n*) for each experimental group/condition, given as a discrete number and unit of measurement |
| ☐ | ☒ | A statement on whether measurements were taken from distinct samples or whether the same sample was measured repeatedly |
| ☒ | ☐ | The statistical test(s) used AND whether they are one- or two-sided *Only common tests should be described solely by name; describe more complex techniques in the Methods section.* |
| ☒ | ☐ | A description of all covariates tested |
| ☐ | ☒ | A description of any assumptions or corrections, such as tests of normality and adjustment for multiple comparisons |
| ☒ | ☐ | A full description of the statistical parameters including central tendency (e.g. means) or other basic estimates (e.g. regression coefficient) AND variation (e.g. standard deviation) or associated estimates of uncertainty (e.g. confidence intervals) |
| ☒ | ☐ | For null hypothesis testing, the test statistic (e.g. *F*, *t*, *r*) with confidence intervals, effect sizes, degrees of freedom and *P* value noted *Give P values as exact values whenever suitable.* |
| ☒ | ☐ | For Bayesian analysis, information on the choice of priors and Markov chain Monte Carlo settings |
| ☒ | ☐ | For hierarchical and complex designs, identification of the appropriate level for tests and full reporting of outcomes |
| ☒ | ☐ | Estimates of effect sizes (e.g. Cohen's *d*, Pearson's *r*), indicating how they were calculated |

*Our web collection on statistics for biologists contains articles on many of the points above.*

## Software and code

Policy information about availability of computer code

| Data collection | SerialEM 3.8.16 |
|---|---|
| Data analysis | cryoSPARC v2 and V3, USF Chimera 1.14, ChimeraX 1.2, COOT 0.8.9.2, Prism 8.0.1, Jalview 2.10.4b1, Phenix 1.18.2, CASTp 3.0, PISA 1.52, Consurf 2016, I-TASSER 5.1, MOLEonline 2.5, Muscle 3.8.31, Psipred 4.02 |

For manuscripts utilizing custom algorithms or software that are central to the research but not yet described in published literature, software must be made available to editors and reviewers. We strongly encourage code deposition in a community repository (e.g. GitHub). See the Nature Portfolio guidelines for submitting code & software for further information.

## Data

Policy information about availability of data

All manuscripts must include a data availability statement. This statement should provide the following information, where applicable:
- Accession codes, unique identifiers, or web links for publicly available datasets
- A description of any restrictions on data availability
- For clinical datasets or third party data, please ensure that the statement adheres to our policy

Structural models of NTCPEM-Np87 and NTCPEM-Nb91 complexes have been deposited in Protein Data Bank (PDB) with accession codes 7PQG and 7PQQ, respectively, and the corresponding cryo-EM maps were deposited in the Electron Microscopy Data Bank (EMDB) under accession numbers EMD-13593, and EMD-13596.

# Field-specific reporting

Please select the one below that is the best fit for your research. If you are not sure, read the appropriate sections before making your selection.

☒ Life sciences ☐ Behavioural & social sciences ☐ Ecological, evolutionary & environmental sciences

For a reference copy of the document with all sections, see nature.com/documents/nr-reporting-summary-flat.pdf

# Life sciences study design

All studies must disclose on these points even when the disclosure is negative.

| | |
|---|---|
| Sample size | No statistical methods were used to predetermine sample size. Functional experiments were done in at least n=3 biologically independent experiments. Sizes of the cryo-EM data sets were predetermined based on microscope availability, and data collected was sufficient to obtain maps with the reported resolutions that enable structure determination. |
| Data exclusions | No functional data was excluded from the analyses. Poor-quality cryo-EM micrographs and particles were removed using standard analysis software and protocols |
| Replication | Functional studies were repeated during biologically experiments as stated in Figure legends and Methods, and results were replicated reasonably well within the experimental error reported. For structural studies, the quality of purified protein and EM specimen was fully reproducible using at least three independent biological replicates |
| Randomization | Randomization is not relevant to this study |
| Blinding | Blinding was not performed, because grouping was not applicable to this study |

# Reporting for specific materials, systems and methods

We require information from authors about some types of materials, experimental systems and methods used in many studies. Here, indicate whether each material, system or method listed is relevant to your study. If you are not sure if a list item applies to your research, read the appropriate section before selecting a response.

### Materials & experimental systems

| n/a | Involved in the study |
|---|---|
| ☐ | ☒ Antibodies |
| ☐ | ☒ Eukaryotic cell lines |
| ☒ | ☐ Palaeontology and archaeology |
| ☒ | ☐ Animals and other organisms |
| ☒ | ☐ Human research participants |
| ☒ | ☐ Clinical data |
| ☒ | ☐ Dual use research of concern |

### Methods

| n/a | Involved in the study |
|---|---|
| ☒ | ☐ ChIP-seq |
| ☒ | ☐ Flow cytometry |
| ☒ | ☐ MRI-based neuroimaging |

## Antibodies

| | |
|---|---|
| Antibodies used | llama (Lama glama) antibody fragments against NTCPCO (nanobodies or VHH) were created and used in this study |
| Validation | Nanobody 87 and 91 binding to NTCPEM was assayed by size-exclusion chromatography in detergent solutions, as well as by fluorescence-based methods in HEK293 cells expressing the transporters as shown and described in Fig. 1b, and Fig. 4c of this study. Binding of Nb87 to human NTCPWT was also confirmed in HEK293 cells as shown and described in Fig. 4c |

## Eukaryotic cell lines

Policy information about cell lines

| | |
|---|---|
| Cell line source(s) | HEK293F (Thermo Fisher) |
| Authentication | No cell-line authentication was performed |
| Mycoplasma contamination | No mycoplasma contamination tests were performed |
| Commonly misidentified lines (See ICLAC register) | No commonly misidentified cell lines were used |

