## [Peer Review File · Nature]

Manuscript Title: Structural basis of sodium-dependent bile salt uptake into the liver

Reviewer Comments & Author Rebuttals

Reviewer Reports on the Initial Version:

Referee #1

The manuscript by Goutam and colleagues reports two structures of the human N⁺-taurocholate co-transporting polypeptide (NTCP) in complexes with a nanobody (Nb87) and a megabody (Mb91), determined using cryo-electron microscopy. While the NTCP-Nb87 structure showed the transporter in its inward-facing state, the complex with Mb91 revealed a pore open to both sides of the membrane. A density found in the middle of the pore was proposed to be a substrate, although the density was not well-defined enough to allow for its unambiguous identification. Existing mutagenesis data agrees with such a location for the substrate. The authors further proposed that preS1 binds to the open pore conformation and used binding assays to support their suggested location. Overall, the work represents a significant advance in understanding the molecular mechanism of the bile salts uptake to the cell. The identification of the binding site of preS1 in NTCP may suggest new ways for preventing HBV infection.

The authors admitted that the open pore structure observed in the NTCP-Mb91 complex was highly unexpected. However, the substrate seen in the middle of the pore suggests that this state may be a transition state between the outward-facing and inward-facing conformations. Perhaps, instead of "open-pore," it can be called a transition conformation.

Some minor comments:

1. Page 8, paragraph 2, change "NTCP-Nb91" to "NTCP-Mb91".
2. ED Figure 3 is of inadequate quality and needs to be remade.
3. Some key information is missing in ED Table 1, such as the magnification for imaging, the imaging mode, the media used for preserving the protein for grid freezing (detergent, nanodisc), the unit for the sharpening B-factor, the B-factor of the refined model.
4. Numbers in the text, table and figures are not written in a consistent way. For example, in ED Figure 2, the number of particles picked is written as 6,535,687, but the number of micrographs is written as 21792. Why include "," in one but not in the other?

Referee #2

Goutam and coworkers report the cryo-EM structure of the hepatocellular sodium-dependent bile acid transporter NTCP (SLC10A1) and show data on molecular details of the transport process. To this end, the authors expressed a tagged and consensus sequence optimized human NTCP in HEK-293F cells for protein isolation. In addition, the authors generated nanobodies and megabodies against NTCP. Purified NTCP was complexed with nanobodies or megabodies and used for cryo-EM structure determination. Functionally, recombinant human NTCP was characterized by overexpression in HEK-293F cells using a fluorescent taurocholate analogue as transport substrate. The authors demonstrate that their NTCP is functional as transporter for the substrate used and

binds GFP-tagged myr-PreS. The structures of NTCP complexed with Mb91 (refined to 3.2 Å) or Nb-87 Fab fragment (refined to 3.7 Å) show an open-pore and an inward-facing state, respectively. The structural and binding data presented are not compatible with an alternate-access transport model but in favor of a gated-pore mechanism.

Comments

1. Abstract: The authors might mention the resolution, which was obtained in this study.
2. Line 35: What is meant by "adsorption"?
3. Line 42: Reference 10 specifically excludes NTCP from involvement of the compounds tested and reference 11 does the same for NTCP. I suggest rephrasing this sentence accordingly or providing different references.
4. Line 73: The authors should give the complete sequence of NTCPCO finally used and align it with NTCPWT.
5. Line 76: A reference for TA-DB should be given. I suggest giving in legend to figure 1 the uptake time for transport.
6. Comment to lines 198 ff: Intracellular (free) bile salt concentrations in hepatocytes are not known exactly, but they are most likely very low: PMID: 8978363, PMID: 7977756. This would reduce efflux along an electrochemical in-to-out gradient.
7. Line 235: Which vertebrate orthologues were used?
8. Line 245: Should "gene" read "cDNA"?
9. Line 293: How were the cells centrifuged (g-force) and was the cell pellet rinsed with substrate free buffer or directly used for measurement?
10. Legends to Figure 3b and figure 4b: The colors (green and purple) should be explained.

Author Rebuttals to Initial Comments:

Referee #1 (Remarks to the Author):

The manuscript by Goutam and colleagues reports two structures of the human N⁺-taurocholate co-transporting polypeptide (NTCP) in complexes with a nanobody (Nb87) and a megabody (Mb91), determined using cryo-electron microscopy. While the NTCP-Nb87 structure showed the transporter in its inward-facing state, the complex with Mb91 revealed a pore open to both sides of the membrane. A density found in the middle of the pore was proposed to be a substrate, although the density was not well-defined enough to allow for its unambiguous identification. Existing mutagenesis data agrees with such a location for the substrate. The authors further proposed that preS1 binds to the open pore conformation and used binding assays to support their suggested location. Overall, the work represents a significant advance in understanding the molecular mechanism of the bile salts uptake to the cell. The identification of the binding site of preS1 in NTCP may suggest new ways for preventing HBV infection.

We thank the referee for the accurate summary and the appreciation that our work is a significant step forward in understanding the molecular mechanisms of bile salt cellular uptake.

The authors admitted that the open pore structure observed in the NTCP-Mb91 complex was highly unexpected. However, the substrate seen in the middle of the pore suggests that this state may be a transition state between the outward-facing and inward-facing conformations. Perhaps, instead of “open-pore,” it can be called a transition conformation.

The Referee is likely right, and the open-pore state is a transition state between outward-facing and inward-facing states, as discussed in the manuscript, but we don't have structural proof of NTCP outward-facing states. Indeed, this is an important mechanistic question to be resolved. We still think it is worth it to highlight that one of the uncovered states displays an open pore, and would like to keep that name, but we edited the second paragraph in the discussion to highlight that it is likely an intermediate and transient state: “...*raising the important question on how to reconcile an open-pore intermediate state with thermodynamically active transport. Our structures suggest a plausible mechanism, whereby the pore is transiently opened in the presence of substrate....*”

Some minor comments:

1. Page 8, paragraph 2, change “NTCP-Nb91” to “NTCP-Mb91”.

The structure referred to on page 8/paragraph 2 is actually that of NTCP-Nb91 complex in nanodisc, which adopts a nearly identical conformation to that of NTCP-Mb91 complex in detergent, within the limits of the reported resolutions.

2. ED Figure 3 is of inadequate quality and needs to be remade.

Thanks for pointing this out, we have remade this figure to improve the quality.

3. Some key information is missing in ED Table 1, such as the magnification for imaging, the imaging mode, the media used for preserving the protein for grid freezing (detergent, nanodisc), the unit for the sharpening B-factor, the B-factor of the refined model.

We apologize for the missing information, and the table has been corrected.

4. Numbers in the text, table and figures are not written in a consistent way. For example, in ED Figure 2, the number of particles picked is written as 6,535,687, but the number of micrographs is written as 21792. Why include “,” in one but not in the other?

Thanks for highlighting this inconsistency. It has been corrected.

Referee #2 (Remarks to the Author):

Goutam and coworkers report the cryo-EM structure of the hepatocellular sodium-dependent bile acid transporter NTCP (SLC10A1) and show data on molecular details of the transport process. To this end, the authors expressed a tagged and consensus sequence optimized human NTCP in HEK-293F cells for protein isolation. In addition, the authors generated nanobodies and megabodies against NTCP. Purified NTCP was complexed with nanobodies or megabodies and used for cryo-EM structure determination. Functionally, recombinant human NTCP was characterized by

overexpression in HEK-293F cells using a fluorescent taurocholate analogue as transport substrate. The authors demonstrate that their NTCP is functional as transporter for the substrate used and binds GFP-tagged myr-PreS. The structures of NTCP complexed with Mb91 (refined to 3.2 Å) or Nb-87 Fab fragment (refined to 3.7 Å) show an open-pore and an inward-facing state, respectively. The structural and binding data presented are not compatible with an alternate-access transport model but in favor of a gated-pore mechanism. We thank the Referee for the detailed summary, as well as the constructive comments on the manuscript and references within.

Comments

1. Abstract: The authors might mention the resolution, which was obtained in this study.

We think it is more appropriate to state the resolution of cryo-EM maps in the text, along with reference to Extended Data Figures showing details of single-particle analysis and maps.

2. Line 35: What is meant by "adsorption"?

Thanks for highlighting the lack of clarity here. We have edited that sentence to: "...shuttling between intestine and liver, where BSs are used to aid nutrient absorption and generate bile, respectively"

3. Line 42: Reference 10 specifically excludes NTCP from involvement of the compounds tested and reference 11 does the same for NTCP. I suggest rephrasing this sentence accordingly or providing different references.

Thanks for pointing out the poor choice of references here. The references have been removed, and we now cite more appropriately:

Ref 10. Tolle-Sander, S., Lentz, K. A., Maeda, D. Y., Coop, A. & Polli, J. E. Increased acyclovir oral bioavailability via a bile acid conjugate. *Mol Pharm* **1**, 40-48, doi:10.1021/mp034010t (2004).

Ref 11. Kullak-Ublick, G. A. *et al.* Chlorambucil-taurocholate is transported by bile acid carriers expressed in human hepatocellular carcinomas. *Gastroenterology* **113**, 1295-1305, doi:10.1053/gast.1997.v113.pm9322525 (1997).

Ref 12. Bhat, L., Jandeleit, B., Dias, T. M., Moors, T. L. & Gallop, M. A. Synthesis and biological evaluation of novel steroidal pyrazoles as substrates for bile acid transporters. *Bioorg Med Chem Lett* **15**, 85-87, doi:10.1016/j.bmcl.2004.10.027 (2005)

4. Line 73: The authors should give the complete sequence of NTCP_{CO} finally used and align it with NTCP_{WT}.

We agree with the Referee that this is important, and have made several changes in the manuscript to improve clarity: 1-we re-named the NTCP construct used for cryo-EM analysis to NTCP_{EM} throughout the text; 2- we made a new Extended Data Figure (revised Extended Data Fig. 1) showing the amino acid sequence alignment of all NTCP constructs used in this work (NTCP_{WT}, NTCP_{CO}, and NTCP_{EM}). In this figure, we also provide the protein sequence IDs of the vertebrate orthologs used during consensus amino acid determination, as the Referee suggested in point 7; 3-We edited the methods section to provide more details on how we built the constructs.

5. Line 76: A reference for TA-DB should be given. I suggest giving in legend to figure 1 the uptake time for transport.

Thanks for pointing this out, we regret the lack of reference to the fluorescent substrate analog in the first version of the manuscript. We have added two references to articles showing the synthesis of the compound (Yamaguchi et al., *Drug. Metab. Pharmacokinet.* 2010), and its use as NTCP substrate (De Bruyn et al., *J. Pharm.* 2014), respectively, in Methods.

6. Comment to lines 198 ff: Intracellular (free) bile salt concentrations in hepatocytes are not known exactly, but they are most likely very low: PMID: 8978363, PMID: 7977756. This would reduce efflux along an electrochemical in-to-out gradient.

Thanks for the interesting remark and the associated references.

7. Line 235: Which vertebrate orthologues were used?

A phylogenetic tree with SLC10A1 vertebrate orthologues, as well as NCBI reference sequence IDs, used to determine consensus amino acids are given in revised Extended Data Fig. 1.

8. Line 245: Should "gene" read "cDNA"?

Thanks for pointing this out, it has been corrected

9. Line 293: How were the cells centrifuged (g-force) and was the cell pellet rinsed with substrate free buffer or directly used for measurement?

In the revised Methods, we provide details on how substrate transport, and myr-PreS1-GFP binding assays were performed, including centrifugation g-forces and washing steps with fluorescent-probe free buffers.

10. Legends to Figure 3b and figure 4b: The colors (green and purple) should be explained.

The figure legends have been edited to describe in more detail the color-code used.